# Improving Physical and Chemical Properties of Saline Soils with Fly Ash Saline and Alkaline Amendment Materials

Changcong An [1], Fenglan Han [1,2,*], Ning Li [1], Jintao Zheng [3], Maohui Li [1], Yanan Liu [1] and Haipeng Liu [1]

1   Institute of Materials Science and Engineering, North Minzu University, Yinchuan 750021, China; zhiqiang7261@163.com (C.A.)
2   International Scientific & Technological Cooperation Base of Industrial Waste Recycling and Advanced Materials, Yinchuan 750021, China
3   Shanxi Shanbei Qianyuan Energy and Chemical Co., Ltd., Baoji 721000, China
*   Correspondence: 2002074@nmu.edu.cn

**Abstract:** Studies have demonstrated that the physicochemical properties of saline soils can be improved, and crop growth can be promoted by fly ash saline and alkaline soil amendment materials. Herein, the effects of fly ash saline and alkaline soil amendment materials on the physical and chemical properties of saline soil and growth conditions of Arrhenatherum elatius at room temperature were evaluated. Meanwhile, planting experiments of *Zea mays* L. were conducted in the demonstration field of saline–alkaline land amendment in Yinchuan, Ningxia. The results showed that the application of amendment materials significantly ($p < 0.05$) improved saline soil's physical and chemical properties. The saline soil pH decreased from an average of 10.51 to 8.89; the $Na^+$ content decreased from an average of 2.93 g·kg$^{-1}$ to 0.7 g·kg$^{-1}$ after 25 days of action. In addition, the soil bulk density decreased from an average of 1.49 g·cm$^{-3}$ to 1.36 g·cm$^{-3}$, and the total porosity increased by 15.60%. Soil available phosphorus and available potassium content also increased significantly, with mean values increasing from 6.74 mg·kg$^{-1}$ and 173 mg·kg$^{-1}$ to 58.30 mg·kg$^{-1}$ and 330.76 mg·kg$^{-1}$, respectively. In addition, the plant height and stem thickness of *Arrhenatherum elatius* increased from an average of 11.76 cm, 1.28 mm to 21.72 cm, 1.59 mm with the application of 2.5 wt% amendment material. The plant height and stem thickness of *Zea mays* L. increased from mean values of 210 cm and 21.94 mm to 315.7 cm and 26.75 mm, respectively, when 0.07 t·hm$^{-2}$ of amendment material was applied in the field. Overall, it was concluded that applying fly ash saline and alkaline soil amendment materials improves the physicochemical properties of saline soils, reducing saline stress and promoting the growth of *Arrhenatherum elatius* and *Zea mays* L.

**Keywords:** circulating fluidized bed; fly ash; fly ash saline amendment materials; saline soil; plants

## 1. Introduction

Soil salinization is a serious obstacle to national food production and sustainable regional economic development [1]. More than 900 million hectares of arable land worldwide is affected by soil salinization [2]. Global salinized soil area accounts for 25% of the total land area [3]. It is distributed in all climatic zones across more than 100 countries and territories. Moreover, the area of salinized soil is gradually increasing, with an annual growth rate of more than 1.5 million hectares [4]. The contradiction between population and land has become obvious further with economic development. Saline soils are an important reserve land resource. Reasonable utilization and amendment of saline soils are crucial to guarantee national food security, protect ecological agriculture, and promote regional economic development [5].

Soil degradation due to salinity and alkalinity is one of the most important obstacles to agricultural production worldwide [6]. The area of saline soil in China reaches $3.69 \times 10^7$ hm$^2$, close to 4.88% of the available land area [7]. It is mainly distributed in arid and semi-arid areas with arid climate, low precipitation, high water table and high soil

evaporation. Examples include the plains in northeast China, Ningxia, Xinjiang and Gansu in northwestern China [8]. Salt stress faced by saline soils, such as excessive $Na^+$ accumulation and high exchangeable sodium ion content, results in plant root ion toxicity, osmotic stress, and metabolic disorders, as well as reduces soil fertility and affects soil hydraulic properties [9]. Alkaline stress significantly increases soil pH, changes physical and chemical properties and structure, weakens soil aeration, and severely hinders crop root growth, affecting seed germination and photosynthesis [10]. Soil salinization reduces crop yields by 18–40%, posing a serious threat to China's food security [11]. At present, there are four primary measures for saline land improvement. One is hydrological improvement, such as drip and diffuse irrigation for salt washing [12]; another is physical improvement, such as sand mixing, straw mulching, and conservation tillage [13,14]. There are also biological improvements, such as planting salt-tolerant plants [15], and chemical improvements, such as the application of desulphurized gypsum, fly ash, and biochar [16]. However, chemical conditioning has been used for a long time in the reclamation of saline or alkaline soils [17].

Globally, coal is still the main fuel for power generation [18]. Coal-fired power generation accounts for 36.4% of the world's energy share [19]. China's economy is undergoing a period of rapid development and there is an increased energy demand. China's proven coal reserves account for 12.84% of the global reserves, and coal will remain the country's main energy source for a long time [20]. Among these reserves, fly ash is the main residual product after coal-fired power generation [21]. However, its low bulk density and the presence of calcium and iron aluminum trioxide make it a potential alternative to gypsum for ameliorating degraded sodic soils. Its annual output in China is more than 600 million tons [22], which has become one of the largest industrial solid wastes generated. Geographical differences in production and marketing and the imbalance of supply and marketing conditions result in the accumulation of a large amount of fly ash. Improper disposal can jeopardize human health and cause serious pollution to the environment [23]. Therefore, the rational application of fly ash is vital for both resource recovery and environmental protection. With the development of the industry, the main application in the field of traditional building materials cannot completely eliminate the stockpiled fly ash, and a more efficient way of fly ash elimination has to be developed urgently. Fly ash has the natural advantage of improving the soil because its main components are similar to those of the soil, and it also contains medium elements, such as P, K, Na, Mg, and trace elements, such as B, Mn, and Zn [24]. So, it is often used in the field of soil amendment. For example, the application of 5–10% fly ash in acidic soils promotes the growth of *Acacia mangium* [25]. The application of fly ash-based soil conditioner significantly promotes the growth of wheat seedlings [26]. In addition, the use of conditioners made from high-iron fly ash effectively increases soil aggregate size and permeability [27] and results in improved water-holding capacity of the soil [28]. In terms of amelioration of saline soils, it is found that application of fly ash significantly reduces the pH of coastal saline soils and increases the soil quick-acting N, P, K content [29]. Soil amendments made with fly ash + vinegar zap + sewage sludge can reduce the sodium adsorption ratio of saline soils and promote the growth of oats [7].

The application of fly ash as a building and high-value-added recycling material has been previously reported [30–33]. In addition, several research works demonstrate the use of fly ash in agriculture as fertilizer to promote crop growth, acidic soils, and to remediate heavy metal pollution [34,35]. However, the changing law of the physical and chemical properties of saline soil after applying fly ash for the amendment of saline soil has been rarely reported. Here, the efficacy of fly ash saline and alkaline soil amendment materials to improve saline soils and their effects on *Arrhenatherum elatius* is discussed. The hypothesis is based on the fact that applying fly ash saline and alkaline amendment materials could significantly improve the physical and chemical properties of saline soils, such as bulk density and pH, thereby facilitating plant growth. The study aims to (a) evaluate the potential heavy metal pollution risk of fly ash saline and alkaline soil amendment materials, (b) evaluate the effect of fly ash saline and alkaline soil amendment materials on the physical and chemical properties of saline soils, and (c) assess the effects of fly ash saline

and alkaline soil amendment materials on *Arrhenatherum elatius* plant height, stem thickness, dry weight, and chlorophyll content, as well as *Zea mays* L. plant height, stem thickness, and bending strength.

## 2. Materials and Methods

### 2.1. Experimental Materials

Circulating fluidized bed fly ash was used as the matrix material to produce fly ash saline and alkaline soil amendment materials. It was provided by a power plant in Ningxia and was supplemented with additives such as MX (large particles of nutrient additives), NF (organic matter), JZ (an acidic organic matter), and CMC-B (water-retaining agent), in addition to other activation additives (a kind of acid modifier). The basic properties of the amendment materials are shown in Tables 1 and 2. Saline–alkaline soils were obtained from the saline–alkaline land amendment demonstration field at Yinchuan City of Ningxia Hui Autonomous Region. The basic physical and chemical properties of the soil for testing are shown in Tables 3 and 4.

**Table 1.** Basic properties of fly ash saline and alkaline soil amendment materials.

| Properties | pH | EC/($\mu s \cdot cm^{-1}$) | Ca% | Mg% | Si% | P% | K% |
|---|---|---|---|---|---|---|---|
| Content | 8.9 | 900.00 | 2.77 | 0.41 | 0.36 | 0.35 | 0.35 |

**Table 2.** Heavy metal content of fly ash saline and alkaline soil amendment materials.

| Properties | Cu ($mg \cdot kg^{-1}$) | Ni ($mg \cdot kg^{-1}$) | Cr ($mg \cdot kg^{-1}$) | Cd ($mg \cdot kg^{-1}$) | Pb ($mg \cdot kg^{-1}$) |
|---|---|---|---|---|---|
| Content | 42.00 | 28.00 | 48.00 | 0.50 | 63.00 |

**Table 3.** Nutrient content of test soil.

| Properties | Organic Matter /% | Available Phosphorus /($mg \cdot kg^{-1}$) | Available Potassium /($mg \cdot kg^{-1}$) | Exchangeable Sodium /($cmol \cdot kg^{-1}$) | Exchangeable Calcium /($cmol \cdot kg^{-1}$) |
|---|---|---|---|---|---|
| Content | 7.10 | 6.74 | 173.00 | 1.32 | 1.03 |

**Table 4.** Basic properties of test soil.

| Properties | pH | EC /($ms \cdot cm^{-1}$) | Bulk Density /($g \cdot cm^{-3}$) | Total Soil Porosity /% | Sodium Ion /($cmol \cdot kg^{-1}$) | Calcium Ion /($cmol \cdot kg^{-1}$) |
|---|---|---|---|---|---|---|
| Content | 10.51 | 2.31 | 1.49 | 40.29 | 2.93 | 0.53 |

### 2.2. Experimental Design

2.2.1. The Effect of Fly Ash Saline and Alkaline Soil Amendment Materials on Soil Properties

The experiments were conducted in April 2023 at room temperature in the Polymer Multifunctional Laboratory, Institute of Materials Science and Engineering, North Minzu University. The mass percentage of amendment materials in 1 kg of saline soil was set at 0 wt%, 0.5 wt%, 1.5 wt%, 2.5 wt%, 5 wt%, 7.5 wt%, and 10 wt% (wt% is weight percent) in seven gradients, respectively (Figure 1). It was mixed well in plastic pots with a top diameter of 14.5 cm, a bottom diameter of 11.5 cm, and a height of 11.5 cm. Water was supplied every two days to maintain 80% of the water-holding capacity in the field, with irrigation water pH of 8.26 and EC of 660 $\mu s \cdot cm^{-1}$, and each gradient was replicated nine times for a total of 63 pots. The experiment was conducted using three different action times (5 d, 15 d, and 25 d). A ring knife was used to measure the soil's physical properties at each action time node, and 300 g of fresh soil samples were taken from each pot at each

action time node, air-dried indoors and passed through a 2 mm pore size sieve to determine the physicochemical indices.

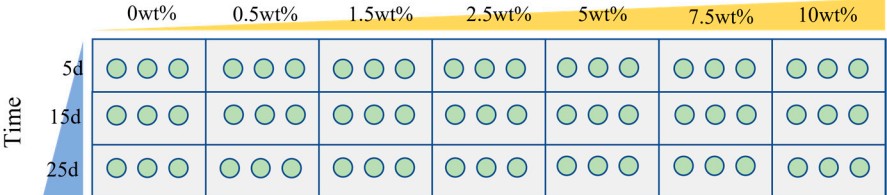

**Figure 1.** Experimental design of the effect of fly ash saline and alkaline soil amendment materials on soil properties. Blue shading represents increasing the duration of action. Yellow shading represents increasing dosages of amendment materials. Green circles represent flower pots.

### 2.2.2. The Effect of Fly Ash Saline and Alkaline Soil Amendment Materials on Plant Growth

For further investigation of the effect of fly ash saline and alkaline soil amendment materials on plant growth, a pot experiment was carried out with seven dosages of fly ash saline and alkaline soil amendment materials. The experiments were conducted at room temperature in the Institute of Materials Science and Engineering, North Minzu University in April 2023. Different dosages (amendment materials applied at the same rate as in Section 2.2.1) of fly ash saline and alkaline soil amendment materials were mixed into 1 kg of test soil. Each dosage was tested in three replications for a total of 21 pots. Thirty *Arrhenatherum elatius* seeds were uniformly sown in 1–2 cm deep soil per pot without applying any bottom fertilizer and supplemented with water every two days to maintain 80% of the water-holding capacity of the field. In addition, the experimental period was set to 25 d, and relevant agronomic traits of *Arrhenatherum elatius* plants were measured at 25 d.

### 2.2.3. Field Experiments in *Zea mays* L. Planting

Field studies were conducted for *Zea mays* L. planting field experiments from April to October 2023 in the saline–alkaline land amendment demonstration field in Yinchuan, Ningxia, China. Two treatments, i.e., CK (no fly ash saline and alkaline land amendment materials applied) and JMC (fly ash saline and alkaline soil amendment materials applied at 0.07 t·hm$^{-2}$), were carried out, with three replications per gradient and 2.5 hm$^2$ per plot. The experimental plots were ploughed deeply and finely, so that the fly ash saline and alkaline soil amendment materials could be evenly mixed with the 0–20 cm tilled soil. Meanwhile, the *Zea mays* L. seeds were sown in April 2023, and the agronomic traits related to corn plants were measured at the milky stage of the corn in September and harvested in October.

### *2.3. Determination Method*

#### 2.3.1. Leaching Content of Heavy Metal Elements in Fly Ash Saline and Alkaline Soil Amendment Materials in Different pH Leaching Solutions

The leaching behavior of five typical heavy metals (Pb, Cu, Cr, Cd, Ni) from the amendment materials at different pH values was investigated by the horizontal shock method. This was used to simulate the leaching of heavy metals after the amendment materials interacted with different salinized soils. The steps were as follows: NaOH was mixed with ultrapure water at pH 8.5, 9.3, 10.3, and 11.3, to prepare 500 mL of the NaOH leaching agent. The dry weight of 10 g of amendment materials in a 500 mL shaking bottle was weighed in accordance with the liquid–solid ratio of 10:1 (L/kg), and the leaching agent was added. The bottle was tightly capped in the horizontal shaking device. The shaking frequency was adjusted for $110 \pm 10$ times/min. Shaking was performed at room temperature 8 h after the removal of the shaking bottle, and kept static for 16 h. The upper layer of the clear liquid was filtered through a 0.45 microporous membrane. Heavy metal concentration was determined by inductively coupled plasma mass spectrometry.

2.3.2. Hakanson Ecological Risk Assessment Method

The Hakanson Ecological Hazard Index (RI) evaluation method is based on the characterization of heavy metals and their environmental behavior, integrating the heavy metal content and its ecological and environmental effects. The ecological hazard index method includes a single pollution factor, a corresponding heavy metal toxicity factor, and a single potential ecological hazard factor, calculated as follows [36]:

$$C_f^i = C_s^i / C_n^i \tag{1}$$

$$E_r^i = T_r^i \times C_f^i \tag{2}$$

$$RI = \sum E_r^i \tag{3}$$

where $C_f^i$ is the pollution coefficient of a heavy metal, $C_s^i$ is the measured value of heavy metals in fly ash saline land amendment materials, $C_n^i$ is the average value of the main soil environmental chemical background in Ningxia, $T_r^i$ is the toxicity response coefficient of heavy metals, and RI is the comprehensive potential ecological risk index; $E_r^i$ is the single coefficient of potential ecological risk. The toxicity response coefficients of heavy metals were set at Cu = 5, Cr = Ni = 2, and Cd = 20 according to the average value of the main soil environmental chemical background in Ningxia. The relationship between $E_r^i$ and RI and pollution is shown in Table 5.

**Table 5.** Relationship between $E_r^i$ and RI and pollution.

| Hazard Level | $E_r^i$ | RI |
|---|---|---|
| Low ecological hazards I | $E_r^i < 40$ | RI < 150 |
| Medium ecological hazards II | $40 \leq E_r^i < 80$ | $150 \leq RI < 300$ |
| High ecological hazards III | $80 \leq E_r^i < 160$ | $300 \leq RI < 600$ |
| High ecological hazards IV | $160 \leq E_r^i < 320$ | |
| Extremely high ecological hazard V | $E_r^i \geq 320$ | $RI \geq 600$ |

2.3.3. Determination of Soil Physical and Chemical Properties

Soil pH was determined in accordance with the electrode method (soil/water = 1:2.5). Soil electrical conductivity (EC) was also determined in accordance with the electrode method (soil/water = 1:5). Organic matter was determined through the cauterization weight loss method [37]. An ultraviolet spectrophotometer (Carry6000i) was used to determine soil-available phosphorus content and available potassium content. The sodium ion content was determined by a flame photometer and the calcium ion content by EDTA complex titration, respectively. Next, in accordance with the flame photometer method (FP6400A) and the atomic absorption spectrophotometer method (240FS AA), the exchange of sodium ions and calcium ions was determined.

2.3.4. Determination of Soil Maximum Water-Holding Capacity and Soil Moisture Loss Rate

Determination of Soil Maximum Water-Holding Capacity

The maximum soil water-holding capacity experiment was conducted at room temperature. Amendment materials with mass percentages of 0 wt%, 0.5 wt%, 1.5 wt%, 2.5 wt%, 5 wt%, 7.5 wt%, and 10 wt%, respectively, were weighed and mixed well with 200 g of test soil and poured into a PVC tube with an inner diameter of 45 mm and a height of 150 mm. The bottom was sealed with two layers of 200-mesh nylon gauze mesh. Three parallel tests were performed per gradient. Then, 300 mL of pure water was slowly added to the PVC pipe, and the soil was slowly moistened until the water seeped out from the bottom. The top layer of the pipe was sealed with cling film and weighed when no water seeped out

from the bottom of the PVC pipe, and the maximum soil water-holding capacity (MWHC) was calculated as follows:

$$\text{MWHC (\%)} = (W2 - W1) \times 100/200 \tag{4}$$

where MWHC denotes the maximum water-holding capacity of the soil, $W_1$ refers to the initial total mass of the PVC pipe filled with a mixture of soil and amendment material, and $W_2$ represents the weight of the bottom of the PVC pipe in the case of no water seepage.

Determination of Soil Moisture Loss Rate

The soil moisture loss rate experiment was conducted at room temperature. Amendment materials with mass percentages of 0 wt%, 0.5 wt%, 1.5 wt%, 2.5 wt%, 5 wt%, 7.5 wt%, and 10 wt%, respectively, were weighed and mixed well with 200 g of test soil and poured into a PVC tube with an inner diameter of 45 mm and a height of 150 mm. The bottom was sealed with two layers of 200-mesh nylon gauze mesh. Three parallel tests were performed per gradient. Then, 300 mL of pure water was poured to saturate the soil (to obtain the state of the soil when it has the maximum water-holding capacity). The top of the tube was sealed with plastic wrap, and the tube was left at room temperature. The mass was weighed daily, and the observation period was 25 d. Three parallel tests were conducted per gradient, and the soil moisture loss rate (SMLR) was calculated as follows:

$$\text{SMLR (\%)} = (W1 - Wi)/(W1 - W0) \times 100 \tag{5}$$

where SMLR denotes the soil moisture loss rate, $W_0$ refers to the initial total mass of the PVC pipe filled with a mixture of soil and amendment material, $W_1$ represents the weight of the soil when it has a state of maximum water-holding capacity, and Wi is the weight weighed each day (i is the number of days weighed).

2.3.5. Determination of Soil Capacity, Porosity, Capillary Porosity, Non-Capillary Porosity and Three Comparisons

The soil surface was scraped flat and a ring knife was placed on the surface. When pressed, the ring knife entered the soil and was filled with soil. Then, the bottom mesh and the bottom cover were placed. The ring knife was weighed after taking the fresh soil, which was recorded as $W_1$. The weighed ring knife was placed in a flat-bottomed stainless steel basin. Water was added to the upper edge of the ring knife and was kept for 10 h. Then, the knife was taken out and the surface water was dried quickly, which weighed a saturated weight of $W_2$. The ring knife was placed in a 2 mm aperture sieve mesh, left for 12 h, and was weighed and recorded as $W_3$. Again, the ring knife was placed into the oven at 105 °C, and dried until it reached the constant weight. Its dry weight was $W_4$. The original mass of the ring knife was W0. Soil bulk density, soil moisture content, porosity, capillary porosity, and non-capillary porosity were calculated in accordance with the following formulas:

$$\text{Soil bulk density (g·cm}^{-3}) = (W4 - W0)/V \tag{6}$$

$$\text{Total soil porosity (\%)} = (W2 - W4)/V \times 100 \tag{7}$$

$$\text{Soil non-capillary porosity (\%)} = (W2 - W3)/V \times 100 \tag{8}$$

$$\text{Soil capillary porosity (\%)} = (W3 - W4)/V \times 100 \tag{9}$$

$$\text{Soil water content (\%)} = (W1 - W4)/(W4 - W0) \times 100 \tag{10}$$

Calculation of soil tripartite ratio from soil porosity and soil water content [38]:

$$\text{Solid phase (\%)} = 1 - \text{total soil porosity} \tag{11}$$

$$\text{Liquid phase (\%)} = \text{total soil porosity} - \text{soil water content} \tag{12}$$

$$\text{Gas phase (\%)} = 1 - \text{solid phase} - \text{liquid phase} \qquad (13)$$

### 2.3.6. Measurement of Plant Agronomic Traits

Plant height, stem thickness, and dry weight were measured using a plastic straight-edge, digital vernier caliper and analytical balance after 25 d of *Arrhenatherum elatius* planting. Leaf chlorophyll content was measured directly using a chlorophyll meter (Model: TYS-B). Determination of the bending strength of the fourth node of *Zea mays* L. was employed using a stem strength meter (Model: YYD-1) during the milky stage of *Zea mays* L. in September. A tape measure was used to measure *Zea mays* L. plant height and a digital vernier caliper was used to measure *Zea mays* L. stem thickness.

### 2.4. Data Collection and Analysis

In order to investigate the effects of different application gradients of the prepared fly ash saline and alkaline soil amendment materials on the physical and chemical properties of saline soil, *Arrhenatherum elatius* and *Zea mays* L. growth, all the obtained data were further analyzed using Microsoft Office 2021. The analysis of variance (ANOVA) was performed in IBM SPSS 27v, and the differences between the different applied gradients of the amendment materials were tested using Duncan's test ($p < 0.05$).

## 3. Results

### 3.1. Leaching Content of Heavy Metal Elements in Fly Ash Saline and Alkaline Soil Amendment Materials in Different pH Leaching Solutions

The leaching content of Pb at different pH values was lower than the detection limit specified in industry standards. There were significant differences in the leaching contents of Cu and Ni at different pH conditions, and the amount of leaching increased with the increase in pH. The highest leaching levels of elemental Cu and Ni, 8.67 $\mu g \cdot L^{-1}$ and 6.8 $\mu g \cdot L^{-1}$, respectively, were observed at pH 11.3. There was no significant difference in the amount of Cd leached at different pH values. The amount of Cr leached was 0.4 $\mu g \cdot L^{-1}$ at pH 8.5, and the amount of Cr leached increased significantly with the increase in pH of the leach solution at pH 9.3, 10.3 and 11.3 (Table 6). The leaching amounts of five typical heavy metals at different pH levels, from 8.5 to 11.3, were in accordance with China's standards on the quality of agricultural irrigation water.

**Table 6.** Leaching content of heavy metal elements in fly ash saline and alkaline soil amendment materials in different pH leaching solutions.

| pH | Cu/($\mu g \cdot L^{-1}$) | Pb/($\mu g \cdot L^{-1}$) | Ni/($\mu g \cdot L^{-1}$) | Cd/($\mu g \cdot L^{-1}$) | Cr/($\mu g \cdot L^{-1}$) |
|------|---------------------|---------------------|---------------------|---------------------|---------------------|
| 8.5  | 6.56 ± 0.014 c | <0.09 | 6.27 ± 0.007 d | 0.21 ± 0.014 a | 0.40 ± 0.007 b |
| 9.3  | 6.25 ± 0.021 d | <0.09 | 6.42 ± 0.085 c | 0.18 ± 0.014 a | 0.45 ± 0.000 a |
| 10.3 | 7.15 ± 0.028 b | <0.09 | 6.60 ± 0.050 b | 0.20 ± 0.000 a | 0.45 ± 0.014 a |
| 11.3 | 8.67 ± 0.000 a | <0.09 | 6.80 ± 0.042 a | 0.20 ± 0.007 a | 0.44 ± 0.026 a |

As determined by Duncan's test, no significant difference at $p < 0.05$ exists between values in a column containing the same letter within a group. Data are the mean ± SE ($n = 3$).

### 3.2. The Hakanson Potential Ecological Risk Index Evaluation

The leaching of elemental Pb from the amendment materials was below the minimum detection limit in the leaching solutions at pH 8.5, 9.3, 10.3, and 11.3. Therefore, the leaching risk of Pb was not considered here. The individual potential heavy metal risk indices Eri of Cu, Cr, Cd, and Ni were all much lower than 40. All of them showed low ecological risks to the environment. The individual potential heavy metal risk indices Eri of Cu at 8.5, 9.3, 10.3, and 11.3 were 1.5694, 1.4928, 1.4928, and 1.494, respectively, which were the same as those of the other potential heavy metals. The potential heavy metal hazard indices were 1.5694, 1.4928, 1.7105, and 2.0742, respectively, which were higher than the ecological hazard indices of Cr, Cd, and Ni. The potential ecological hazard indices RI of the four heavy metals at pH 8.5, 9.3, 10.3, 11.3 were 2.0586, 1.9864, 2.2207, 2.5977, respectively,

which were much lower than 150, and the results indicated that they were less ecologically hazardous (Table 7).

**Table 7.** Hakanson potential ecological risk index of fly ash saline and alkaline soil amendment materials under different pH conditions.

| pH | Metals | $C_s^i$ | $C_n^i$ | $C_f^i$ | $T_r^i$ | $E_r^i$ | RI |
|---|---|---|---|---|---|---|---|
| 8.5 | Cu | 6.56 | 20.9 | 0.3139 | 5 | 1.5694 | 2.0568 |
| | Cr | 0.4 | 61.3 | 0.0065 | 2 | 0.0131 | |
| | Cd | 0.21 | 92 | 0.0023 | 20 | 0.0457 | |
| | Ni | 6.26 | 29.2 | 0.2144 | 2 | 0.4288 | |
| 9.3 | Cu | 6.24 | 20.9 | 0.2986 | 5 | 1.4928 | 1.9864 |
| | Cr | 0.45 | 61.3 | 0.0073 | 2 | 0.0147 | |
| | Cd | 0.18 | 92 | 0.0020 | 20 | 0.0391 | |
| | Ni | 6.42 | 29.2 | 0.2199 | 2 | 0.4397 | |
| 10.3 | Cu | 7.15 | 20.9 | 0.3421 | 5 | 1.7105 | 2.2207 |
| | Cr | 0.45 | 61.3 | 0.0073 | 2 | 0.0147 | |
| | Cd | 0.2 | 92 | 0.0022 | 20 | 0.0435 | |
| | Ni | 6.6 | 29.2 | 0.2260 | 2 | 0.4521 | |
| 11.3 | Cu | 8.67 | 20.9 | 0.4148 | 5 | 2.0742 | 2.5977 |
| | Cr | 0.44 | 61.3 | 0.0072 | 2 | 0.0144 | |
| | Cd | 0.2 | 92 | 0.0022 | 20 | 0.0435 | |
| | Ni | 6.8 | 29.2 | 0.2329 | 2 | 0.4658 | |

### 3.3. The Effect of Fly Ash Saline and Alkaline Soil Amendment Materials on Soil pH and EC

For the same duration of action, soil pH decreased significantly with the increase in application rate. After 25 d of action, the pH varied in the range of 8.69–9.97, where the pH of 10 wt% treatment was reduced by 1.29 units compared to 0 wt% treatment (Figure 2a). The differences in soil EC values were not significant for the 15 and 25 d duration of action. For the 5 wt%, 7.5 wt%, and 10 wt% treatments, as the duration of action was increased up to 25 d, the soil EC values were significantly increased compared to those of the low-application group, with a range from 626.5 to 1115.0 µs·cm$^{-1}$ (Figure 2b).

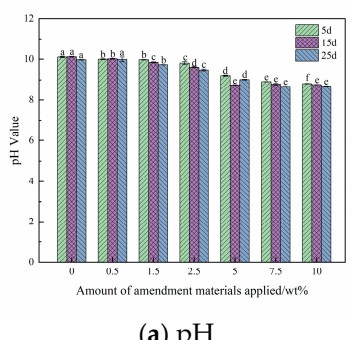

**(a)** pH

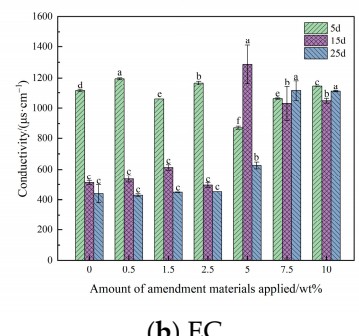

**(b)** EC

**Figure 2.** Changes in pH and EC of saline soil after applying saline and alkaline soil amendment materials. (**a**) Changes in soil pH. (**b**) Changes in soil EC. Columns labeled with the same letter and duration of action are not significantly different at the *p* < 0.05 level. Each mean is accompanied by a standard error (*n* = 3).

### 3.4. The Effect of Fly Ash Saline and Alkaline Soil Amendment Materials on Soil Organic Matter, Available Phosphorus and Available Potassium Content

The soil organic matter content variation ranged from 3.24% to 6.17% for 5 d action time. The soil organic matter content was significantly lower in 5 wt% (3.24%), 7.5 wt% (3.97%), and 10 wt% (4.31%) treatments compared to 0 wt% (5.65%) treatment. At 15 d and 25 d action time, soil organic matter content showed a gradual trend with the increase in the amount of amended material applied, and the variation range of soil organic matter content was 3.19% to 5.89% and 3.66% to 6.32%, respectively (Figure 3a). Soil available phosphorus

content increased significantly after the application of amended materials. The ranges of variation were 13.81–68.88 mg·kg$^{-1}$, 6.30–58.55 mg·kg$^{-1}$, and 10.95–58.30 mg·kg$^{-1}$ at 5 d, 15 d, and 25 d, respectively. The same gradient treatments showed decreased soil available phosphorus content with increasing time, but it was still significantly higher than the 0 wt% treatment (Figure 3b). The changes in soil available potassium content were similar to those of the available phosphorus content. After 25 d of action time, the soil's available potassium content decreased significantly compared to that at 5 d. Still, it was higher than the control 0 wt% treatment, with a range of 150.05–330.76 mg·kg$^{-1}$ (Figure 3c).

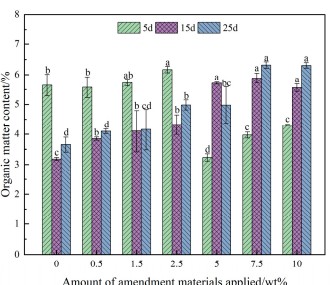 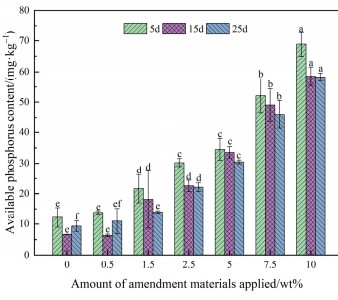 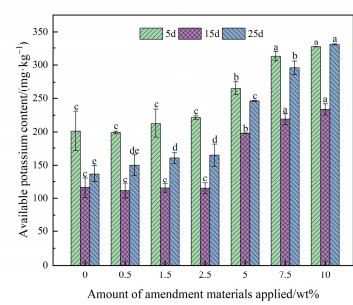

(**a**) Organic matter  (**b**) Available phosphorus  (**c**) Available potassium

**Figure 3.** Changes in nutrient content of saline soil after applying fly ash saline and alkaline soil amendment materials. (**a**) Changes in soil organic matter. (**b**) Changes in soil available phosphorus. (**c**) Changes in soil available potassium. Columns labeled with the same letter and duration of action are not significantly different at the *p* < 0.05 level. Each mean is accompanied by a standard error (*n* = 3).

*3.5. The Effect of Fly Ash Saline and Alkaline Soil Amendment Materials on Soil Sodium Ion, Calcium Ion, Exchangeable Sodium Ion, Exchangeable Calcium Ion Content*

After 5 d action time, there was a significant difference between the treatments in terms of sodium; soil sodium ion content gradually decreased, compared with 0 wt% treatment, in the range of 0.76–1.26 g·kg$^{-1}$. After a 15 d action time, 0.5 wt%, 1.5 wt%, and 2.5 wt% treatments further increased the removal of sodium ions from soil and reached the minimum standard at 25 d; there was no significant difference between the treatments. However, with further increase in application rate, 5 wt%, 7.5 wt%, and 10 wt% treatments showed significant difference in sodium ion content after 15 and 25 d of action, which increased by 87.35%, 48.85% and 56.86%, respectively, compared to 5 d of action (Figure 4a). At 5 d action time, soil calcium ion content was significantly increased in 1.5 wt% (0.38 g·kg$^{-1}$) and 2.5 wt% (0.36 g·kg$^{-1}$) treatments compared to 0 wt% (0.34 g·kg$^{-1}$) one. In contrast, the soil calcium ion content in 5 wt% (0.07 g·kg$^{-1}$), 7.5 wt% (0.16 g·kg$^{-1}$), and 10 wt% (0.18 g·kg$^{-1}$) treatments was significantly reduced compared to 0 wt% one. Calcium ion content ranged from 0.03 to 0.20 g·kg$^{-1}$ and 0.02 to 0.21 g·kg$^{-1}$ after 15 and 25 d of action, respectively. In addition, 0 wt%, 0.5 wt%, 1.5 wt%, and 2.5 wt% treatments showed a significant decrease in the calcium ion content compared to the 5 d action time (Figure 4b).

The 2.5 wt% treatment significantly reduced the soil exchangeable sodium Ion content at each treatment period. After 25 d, the soil exchangeable sodium ion content was significantly reduced by 25.12% in the 2.5 wt% treatment compared to the 0 wt% treatment. After 25 d, the 5 wt%, 7.5 wt%, and 10 wt% treatments significantly increased the soil exchangeable sodium ion content by 99.35%, 34.19% and 80.65%, respectively, compared with the 0 wt% treatment (Figure 4c). After 5 d and 15 d of action, the soil exchangeable calcium ion content of all treatments with amendment materials increased to different degrees compared to the 0 wt% treatment. The variation ranged from 4.84% to 50.00% and 1.93% to 65.22%. After 25 d action time, the 10 wt% treatment showed the most significant increase of 46.55% in exchangeable calcium ion content over the 0 wt% treatment (Figure 4d).

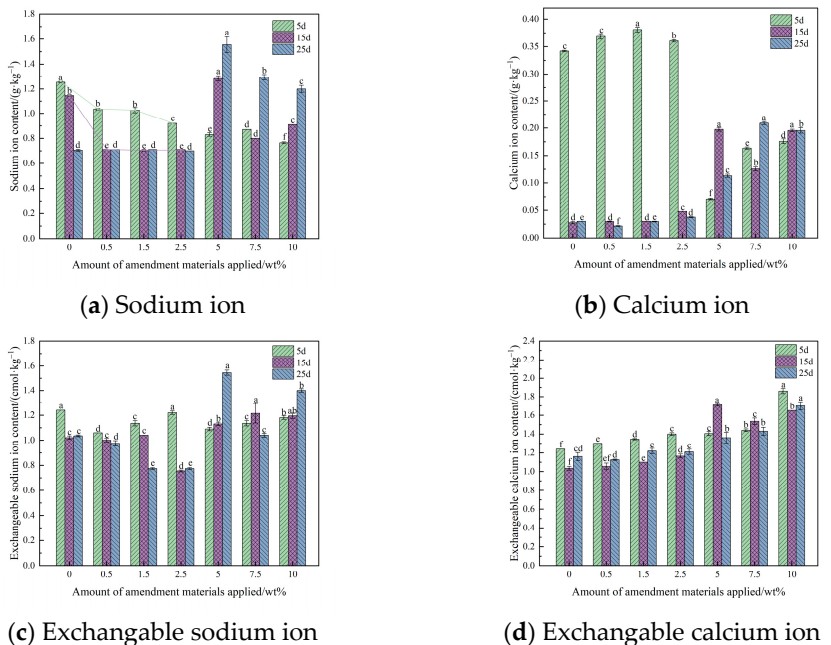

(**a**) Sodium ion

(**b**) Calcium ion

(**c**) Exchangable sodium ion

(**d**) Exchangable calcium ion

**Figure 4.** Changes in chemical properties of saline soil after applying fly ash saline and alkaline soil amendment materials. (**a**) Changes in soil sodium ion. (**b**) Changes in soil calcium ion. (**c**) Changes in soil exchangeable sodium ion. (**d**) Changes in soil exchangeable calcium ion. Columns labeled with the same letter and duration of action are not significantly different at the $p < 0.05$ level. Each mean is accompanied by a standard error ($n = 3$).

### 3.6. The Effect of Fly Ash Saline and Alkaline Soil Amendment Materials on Soil Maximum Soil Water-Holding Capacity and Moisture Loss Rate

The maximum water-holding capacity of the soil increased gradually with the increase in the modified material. The maximum soil water-holding capacity was 32.59%, 32.78%, and 32.97% for 0 wt%, 0.5 wt%, and 1.5 wt% treatments, respectively, with no significant difference. The maximum water-holding capacity of the soil was significantly increased at 2.5 wt%, 5 wt%, and 7.5 wt% treatments compared to 0 wt%, which were 33.58%, 33.78%, and 35.17%, respectively. The maximum soil water-holding capacity in the 10 wt% treatment (39.12%) was significantly higher than that in the 0 wt% treatment (Figure 5a). The addition of improved materials effectively reduced the evaporation of soil water, and at the same time, water evaporation of the treatment group with improved materials was lower than that of the 0 wt% treatment without improved materials. In addition, the 10 wt% treatment soil evaporation rate was minimized at different action times (Figure 5b).

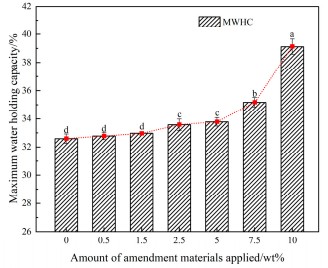
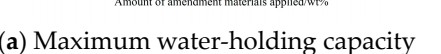
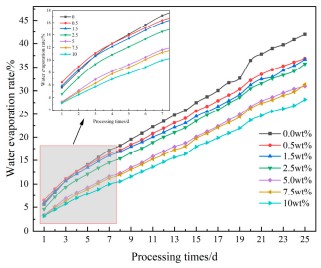

(**a**) Maximum water-holding capacity

(**b**) Water evaporation rate

**Figure 5.** Effect of fly ash saline and alkaline soil amendment materials on maximum water-holding capacity and moisture loss rate of saline soil. (**a**) Changes in soil maximum water-holding capacity. (**b**) Changes in soil water evaporation rate. Columns labeled with the same letter and duration of action are not significantly different at the $p < 0.05$ level. Three biological replicates were performed for each gradient. Bars represent the standard error, from three biological replicates.

### 3.7. The Effect of Fly Ash Saline and Alkaline Soil Amendment Materials on Soil Bulk Density, Total Porosity, Capillary Porosity, and Non-Capillary Porosity

Soil bulk density showed a decreasing and then increasing trend with the increase in amendment material application in the same treatment time. With 2.5 wt% treatment, soil bulk density was significantly reduced compared to 0 wt% treatment by 8.72%, 2.76%, and 6.21% at 5 d, 15 d, and 25 d, respectively. For 25 d action time, soil bulk density was significantly higher in the 5 wt%, 7.5 wt%, and 10 wt% treatments than in the 2.5 wt% treatment, with an increase of 3.73%, 4.35%, and 3.56%, respectively (Figure 6a). Soil porosity varied from 37.29% to 45.04% at 5 d action time, with 5 wt% treatment (45.04%) soil porosity being the largest. The total porosity of the soil ranged from 38.33% to 44.30% at 15 d. The total porosity of the 10 wt% treatment (44.30%) was significantly higher than that of the 0 wt% treatment (40.19%). At 25 d, soil porosity was significantly higher in the 1.5 wt% (45.31%), 2.5 wt% (45.22%), and 5 wt% (46.23%) treatments than in the 0 wt% (39.99%), 0.5 wt% (40.51%), 7.5 wt% (42.21%), and 10 wt% (42.22%) treatments (Figure 6b).

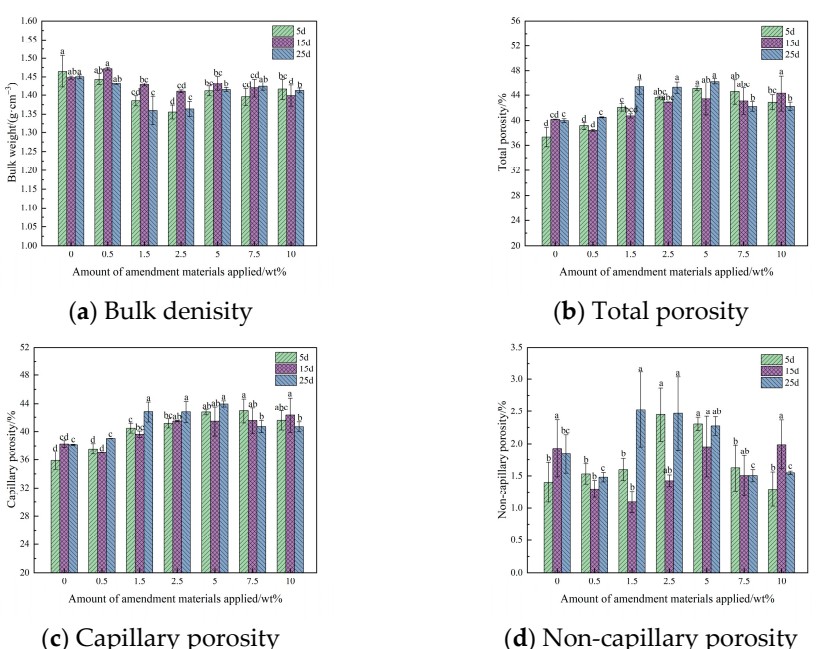

(**a**) Bulk denisity

(**b**) Total porosity

(**c**) Capillary porosity

(**d**) Non-capillary porosity

**Figure 6.** Changes in physical properties of saline soil after applying fly ash saline and alkaline soil amendment materials. (**a**) Changes in soil bulk density. (**b**) Changes in soil total porosity. (**c**) Changes in soil capillary porosity. (**d**) Changes in soil non-capillary porosity. Columns labeled with the same letter and duration of action are not significantly different at the $p < 0.05$ level. Each mean is accompanied by a standard error ($n = 3$).

Soil capillary porosity treatments at 1.5 wt%, 2.5 wt% and 5 wt%, 7.5 wt%, and 10 wt% varied from 40.47% to 42.93% and 39.63% to 42.32% over the 5 d and 15 d action times, respectively. In addition, they were significantly higher than the 0 wt% and 0.5 wt% treatments at 5 d and 15 d. After 25 d of action, soil capillary porosity was significantly higher in the 1.5 wt% (42.78%), 2.5 wt% (42.75%), and 5 wt% (43.96%) treatments than in the 0 wt% (38.15%), 0.5 wt% (39.03%), 7.5 wt% (40.70%), and 10 wt% (40.67%) treatments (Figure 6c). At 5 d, soil non-capillary porosity ranged from 1.29 to 2.45%. The 2.5 wt% (2.45%) and 5 wt% (2.30%) treatments had the largest soil non-capillary porosity. At 15 d, the soil non-capillary porosity varied from 1.09% to 1.99%. In the 0.5 wt% (1.30%) and 1.5 wt% (1.09%) treatments, the non-capillary porosity was significantly reduced compared to 0 wt% (1.93%). After 25 d of action time, the range of non-capillary porosity of the soil varied from 1.48% to 2.53%. In the 1.5 wt% (2.53%) and 2.5 wt% (2.47%) treatments, the non-capillary porosity of the soil was significantly higher than the other treatments (Figure 6d).

### 3.8. The Effect of Fly Ash Saline and Alkaline Soil Amendment Materials on Soil Three Comparisons

The solid phase percentage of the soil was lower than that of the 0 wt% treatment for different applied gradient treatments after a 5 d action time. Among them, the solid phase percentage changed most in the 5 wt% treatment, which was 12.36% lower than that of the 0 wt% treatment. The liquid phase percentage increased in all treatments, ranging from 7.69% to 37.02%, and the gas phase percentage did not change significantly. After 25 d action time, the solid phase percentage of the soil further decreased, ranging from 0.83% to 10.33%, except for the gas phase percentage of the 7.5 wt% treatment, which decreased by 2.99% compared with the 0 wt% treatment, and the liquid phase percentage of the 0.5 wt% treatment, which decreased by 1.85%. All treatments of the liquid and gas phases had different degrees of increasing tendency with changes in the applied amount (Figure 7).

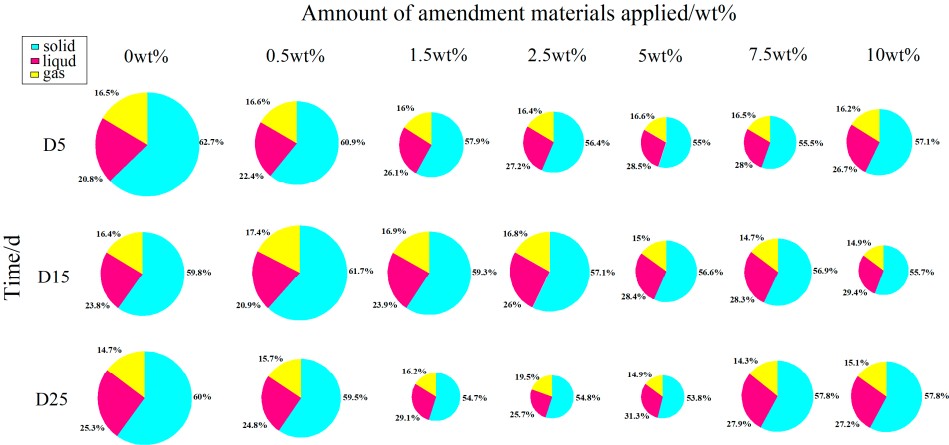

**Figure 7.** Changes in the percentage of solid, liquid, and gas phases of saline soil after applying fly ash saline and alkaline soil amendment materials. The values outside the pie chart represent the percentage of each of the three phases of soil solid, liquid, and gas, and the area of the pie chart represents the percentage of soil solid phase at different gradients of amendment material. Three biological replicates were performed for each gradient. Three biological replicates were performed for each gradient at different times, totaling 63 samples.

### 3.9. The Effect of Fly Ash Saline and Alkaline Soil Amendment Materials on Plant Agronomic Traits

*Arrhenatherum elatius* height varied from 7.03 to 8.03 cm. In 1.5 wt% and 2.5 wt% low-application treatments, *Arrhenatherum elatius* plant height was significantly increased by 39.20% and 84.69% compared to 0 wt% (11.76 cm), respectively. The 5 wt%, 7.5 wt%, and 10 wt% treatments significantly reduced plant height compared to the 0 wt% treatment, with variations ranging from 7.03 to 8.03 cm (Figure 8a). The study likewise found that 1.5 wt% and 2.5 wt% application of amendment materials significantly increased *Arrhenatherum elatius* stem thickness by 20.31% and 24.22%, respectively, compared to 0 wt% (1.28 mm) (Figure 8b). The 1.5 wt% and 2.5 wt% application rates significantly increased the dry weight of *Arrhenatherum elatius* by 6.32% and 60.54%, respectively, compared to 0 wt% (0.1313 g). Meanwhile, in the 5 wt%, 7.5 wt%, and 10 wt% treatments, the dry weight of *Arrhenatherum elatius* was significantly lower compared to that of the 0 wt% treatment, with variations ranging from 0.0915 to 0.0936 g (Figure 8c). The chlorophyll content of *Arrhenatherum elatius* was significantly higher in the 1.5 wt% (24.79 SPAD) and 2.5 wt% (30.46 SPAD) treatments than in the 0 wt% (15.73 SPAD) one. During the 5 wt%, 7.5 wt%, and 10 wt% treatments, the chlorophyll content was reduced, and the chlorophyll content varied from 10.3 to 13.66 SPAD (Figure 8d). The study's results revealed that saline soil pH gradually decreased with the increase in the application amount. This means plant growth was subjected to lesser alkali stress, and the decrease in pH was favourable for plant growth and root uptake of the nutrient content. In this experiment, the application

gradients of 1.5 wt% and 2.5 wt% significantly increased agronomic traits such as plant height, stem thickness, and dry weight of *Arrhenatherum elatius*.

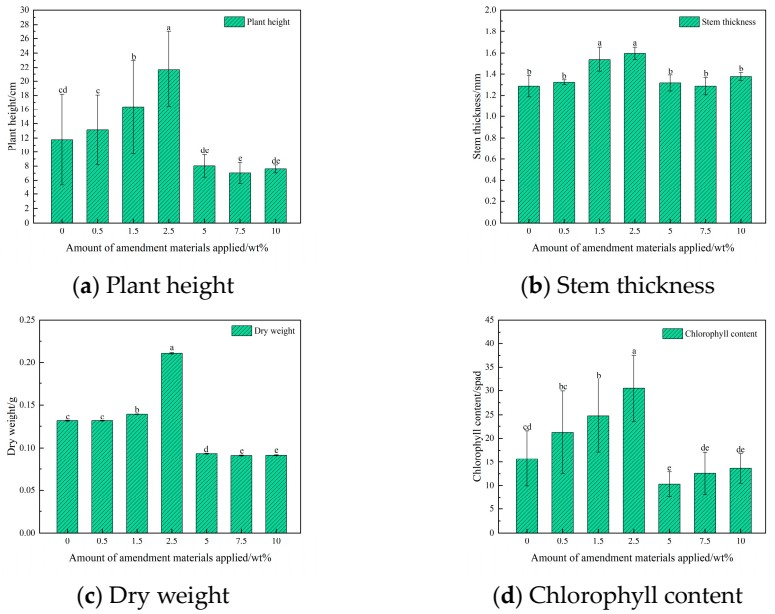

(**a**) Plant height

(**b**) Stem thickness

(**c**) Dry weight

(**d**) Chlorophyll content

**Figure 8.** Effect of fly ash saline and alkaline soil amendment materials on plant growth. (**a**) Changes in plant height. (**b**) Changes in stem thickness. (**c**) Changes in dry weight. (**d**) Changes in chlorophyll content. Columns labeled with the same letter and duration of action are not significantly different at the $p < 0.05$ level. Each mean is accompanied by a standard error ($n$ = 3).

### 3.10. Field Experiment

Field experiments were conducted in the demonstration field of saline–alkaline land amendment in Yinchuan, Ningxia. Plant height, fourth node stem thickness, and bending strength were measured at the milky stage of *Zea mays* L. *Zea mays* L. plant height increased by 50.33% in the amended soil compared to the results obtained after unamended material treatment, stem thickness increased by 21.92%, and bending strength increased by 37.51% (Figure 9a). The seedling emergence rate of *Zea mays* L. in the seedling and nodulation stages of unamended soil was significantly suppressed. In the JMC treatment, the corn seedling emergence rate was significantly improved, as well as the agronomic traits such as plant height and stem thickness (Figure 9b).

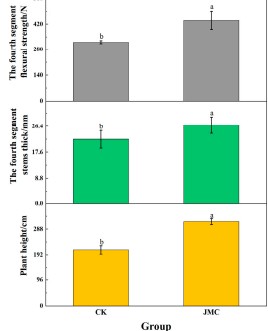
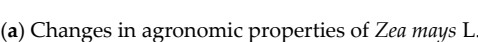
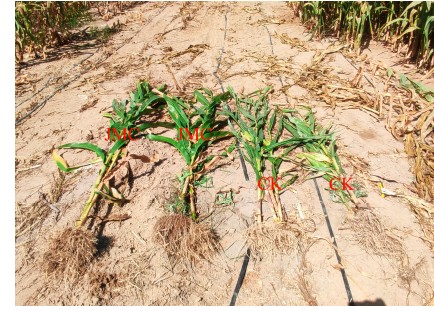

(**a**) Changes in agronomic properties of *Zea mays* L.

(**b**) CK-treated and JMC-treated *Zea mays* L.

**Figure 9.** Changes in agronomic traits related to *Zea mays* L. after the application of fly ash saline and alkaline soil amendment materials. (**a**) Changes in agronomic properties of *Zea mays* L. Columns labeled with the same letter are not significantly different at the $p < 0.05$ level. Each mean is accompanied by a standard error ($n$ = 3); and (**b**) CK-treated and JMC-treated *Zea mays* L. plants in order from right to left.

## 4. Discussion

The Ningxia Hui Autonomous Region features low precipitation and severe water evaporation, and the salts dissolved in water tend to accumulate in the surface layer of the soil, deteriorating the physical structure and chemical properties of the soil and producing soil salinization [39]. In such cases, rapid and effective chemical amendments are essential to alleviate the undesirable properties of the soil and thus cultivate a favourable environment for crop growth [40,41]. Due to its small particle size, rich pore space and specific surface area, fly ash is endowed with the natural advantage of improving the soil structure, and its composition is similar to that of soil, furnishing it with nutrients necessary for plant growth. Applying fly ash saline and alkaline soil amendment materials prepared from circulating fluidized bed fly ash on saline soils significantly improved physical properties such as saline soil bulk weight, porosity, and chemical properties such as pH and exchanged sodium ion content to varying degrees. Meanwhile, the growth of *Arrhenatherum elatius* and *Zea mays* L. was also promoted.

Fly ash has the natural advantage of improving the soil, but fly ash contains some harmful heavy metals. The content of heavy metals in the amendment material and the leaching of heavy metal elements under different pH conditions must meet the limits of national standards, which is a necessary prerequisite for determining whether the improved material can be applied in this field [42]. Our research shows that the content of heavy metals in the amendment materials and the leaching amount at different pH levels have to meet the limits of national standards. This is the essential prerequisite to determine whether the amendment materials can be applied in the field. The contents of the five typical heavy metals (Pb, Cu, Cr, Cd, Ni) in the amendment materials in this study complied with the Chinese standards on the quality of agricultural irrigation water. Of these, the leaching amounts of the amendment materials at different pH levels also complied with the relevant industry standards. These results were consistent with the previous study in which they could not be leached out under alkaline conditions but could be significantly leached out under very acidic conditions, at pH < 4 [5].

In addition, in order to accurately evaluate whether the residual heavy metals in the amendment materials would pose a potential risk of pollution to the environment, the Hakanson risk evaluation method was used to evaluate the potential hazard of heavy metals to the environment [43–45]. The potential ecological hazard indices RI of Cu, Cr, Cd, and Ni in the modified materials were 2.0586, 1.9864, 2.2207, and 2.5977 when the pH levels of the leaching solution were 8.5, 9.3, 10.3, and 11.3, respectively, which were much lower than 150, indicating that they were less ecologically hazardous. Individual potential risk index Eri and potential ecological hazard index EI were both far below the low ecological hazard standard limit. The pH value of the soil in the test area was 10.51, indicating that the amendment material can be safely used for saline soil reclamation.

Carbonates and bicarbonates produced by the hydrolysis of large amounts of exchangeable sodium ions on soil composite colloids are one of the reasons for the high pH of saline soils [46]. This destroys the soil's physical structure by inducing dispersion and spreading of clay particles from the agglomerates [47]. In this regard, an effective method is to apply calcium and magnesium additives, such as desulfurization gypsum, to displace exchangeable sodium ions on soil complex colloids [48]. Our study showed a significant reduction in pH and EC values of saline soils after the application of amendment materials. Where EC is a measure of the total ionic conductivity of the soil, this indicates a decrease in the soluble salt-based ions in the soil, which was conducive to the amendment of the stability of soil aggregates [49].

The high salinity and pH of saline soils inhibit the effectiveness of phosphorus and potassium and prevent the accumulation of soil organic matter [50–52]. Therefore, while reducing salt and draining alkali, a large amount of artificial input of organic matter is necessary to promote soil fertility amendment [53]. Our research found that amendment materials are nutrient-rich. As the application amount increases, nutrient contents such as available phosphorus, available potassium, and organic matter increase significantly in

saline soil. Adding nutrients through exogenous amended materials creates a favorable environment for plant growth by providing necessary nutrients. However, the content of the same applied gradient treatments decreases significantly with the increase in the action time. This is caused by nutrient leaching due to the high-frequency leaching process, thus decreasing the content [54–56].

The mechanism of saline soil amendment by fly ash saline and alkaline soil amendment materials needs to be clarified. Ca and Mg in the amendment materials used in this experiment are 2.77% and 0.41%, respectively. One important reason for the reduction in salinity is the release of calcium in the amendment materials, which displaces the exchangeable sodium ions on the soil composite colloid to the soil solution, reducing the soil exchangeable sodium ion content and increasing the exchangeable calcium ion content, also removiing the sodium ions from the shallow soil by drenching [57–59]. Our experiments found that the 1.5 wt% and 2.5 wt% treatments of low-applied amendment materials significantly accelerated the rate of leaching of sodium ions from the soil compared to the blank control 0 wt% treatment. However, overapplication of the amendment materials failed to reduce soil sodium ion content and increased soil salinity. In addition, lower application gradients significantly reduced soil exchangeable sodium ion content and increased exchangeable calcium ion content. In the 2.5 wt% treatment, soil exchangeable sodium ion content decreased by 25.12% compared with the 0 wt% treatment, and exchangeable calcium ion content increased by 46.55% in the 10 wt% treatment compared with the 0 wt% treatment. In addition, organic matter NF and JZ present in the amendment materials are also direct calcium sources that accelerate sodium removal [60]. The soil calcium ion content of the low-applied amount significantly decreased at 15 d and 25 d of action, which was hypothesized to be because of the low-applied amount itself. Exogenous calcium ion content was low. The high frequency of the leaching process forced the calcium ion to leach from the soil, whereas the exogenous calcium ion introduced into the treatment of a high-applied amount had a larger content. There was no significant decrease in the calcium ion content, even after a long leaching process [61,62].

Yinchuan City of Ningxia Hui Autonomous Region has high average summer temperature, sparse precipitation, and intense soil moisture evaporation, resulting in low water availability to plants. Our research found that applying amendment materials increases the maximum water-holding capacity of the soil and decreases the water loss rate. This provides a good water environment that is required during plant growth. This is because the base material of the improved material, fly ash, has a small particle size with abundant specific surface area and pore space. This increases the soil microporosity after application to the soil, thus increasing the soil's water-holding capacity [24]. Simultaneously, $SiO_2$ and $Al_2O_3$ in the amendment material develop hydration and retain water. In addition, the additive MX is also an essential factor that affects water evaporation; MX contains a large amount of cellulose. Cellulose contains several hydrophilic carboxyl groups with strong flocculation and water retention. This improves water retention and inhibits the evaporation of water [37].

Saline soil has high capacity, low porosity, and a dense structure, which seriously impedes the penetration of plant roots in the soil [63]. In this study, the application of amendment materials can significantly change the three-phase structure of the soil: the proportion of the solid phase is reduced, whereas that of the liquid and gas phases changes to different degrees. Fly ash contains many powder-grade particles, and large-scale applications tend to change soil texture by increasing the powder content [64]. In addition, organic additives such as NF and JZ in the amendment materials provide rich humus that promotes the agglomeration of small soil particles to form micro agglomerates. They increase the stability of the agglomerates and result in the formation of a multistage pore structure. This leads to reduced soil bulk weight, increased porosity, and a recreation of the soil's physical structure [65,66]. Amendment in the soil's physical structure is related to soluble calcium and magnesium in fly ash, which binds fine particles together through cation bridging [67]. However, due to the small particle size of fly ash, a large applied

gradient causes smaller particles to fill in the voids of the soil structure. This results in decreased soil porosity and increased bulk weight.

The high salt content in saline soils is mainly caused by the high-water table and excessively dense capillary porosity, which allow for salts to migrate to the soil surface [68]. Our research found that after applying amendment materials, soil bulk density decreased, total porosity increased, and soil structure became loose. This increased the non-capillary porosity of the soil and, at the same time, cut off the capillary pores, allowing for salts to migrate deeper into the soil through leaching and inhibiting upward salt migration. On the other hand, capillary porosity is related to the size of the adequate water capacity of the soil and the water supply capacity [69]. After applying amendment materials, the number of soil capillary pores and the soil capillary water content increased. Thus, the maximum water-holding capacity of the soil also increased.

Previous studies have shown that soil application of fly ash increases wheat, mustard, and carrot yields [70–72]. However, with further increase in application, seed germination and plant growth were severely inhibited at 5%, 7.5%, and 10% application rates, which could be considered salt stress on plants. Treatments of 5%, 7.5%, and 10% exhibited a significant increase in soil EC, which is a measure of the total ionic conductivity of the soil. When the EC value is too low, it prevents the plant from absorbing nutrients, and when it is too high, it causes seed germination. In addition, plant root growth suffers from severe salt stress and ion toxicity, which inhibits plant growth. Applying fly ash at low concentrations is beneficial as a fertilizer for sustainable crop growth but toxic to crops at high concentrations, consistent with previous studies' findings [73,74]. This is because of the elevated levels of sulphate, chloride, carbonate, and bicarbonate, as well as increased salinity owing to severe nitrogen deficiency [28].

Field experiments showed that *Zea mays* L. plant height, stem thickness, and bending strength significantly increased after applying fly ash saline and alkaline soil amendment materials. *Zea mays* L. resistance to fall significantly improved. These results are also consistent with crop responses to fly ash in other studies [35]. As the amendment material is rich in available phosphorus, available potassium, organic matter, calcium, magnesium, silicon, and other micronutrients, it can provide the necessary nutrient environment for the growth and development of corn. In addition, applying amendment materials can effectively lower saline pH and reduce exchangeable sodium ion content, reducing salt stress on *Zea mays* L. growth. The amendment of soil capacity, porosity, and other physical structures can maintain the balance of water, fertilizer, air, and heat suitable for crop growth, providing a good environment for *Zea mays* L. root growth. Thus, applying fly ash saline and alkaline soil amendment materials significantly reduced saline and alkaline soil pH and exchangeable sodium ion content, increased soil porosity and water-holding capacity, and slowed water evaporation. Farmers in the Ningxia region can alleviate the poor soil structure caused by high salinity and high alkalinity by applying fly ash saline and alkaline soil amendment materials to provide a favourable growing environment and improve the growth of crops.

## 5. Conclusions

Herein, investigations were carried out upon the effects of using fly ash saline soil amendment materials on the physicochemical properties of saline soil as well as plant growth and development. The results showed that after the application of the amendment materials, the pH was effectively reduced and the leaching of sodium ions from the saline soil was accelerated. The physical properties of saline soil such as bulk density and porosity significantly improved. In addition, soil organic matter, available phosphorus, and available potassium were significantly increased as well. The 2.5 wt% gradient treatment significantly promoted *Arrhenatherum elatius* stem thickness and plant height. Still, field application of $0.07 \text{ t·hm}^{-2}$ of amendment material increased *Zea mays* L. plant height, stem thickness, and bending strength. Fly ash saline soil amendment material functioning was proven to show

positive effect on soil and crops, but the effect depends on the sources (difference in coal combustion and sintering temp), land use, weather conditions, etc.

**Author Contributions:** C.A.: Writing—original draft; Data curation; Writing—review and editing. F.H.: Funding acquisition; Conceptualization. J.Z.: Project administration. N.L.: Supervision; Project administration. M.L.: Methodology. Y.L.: Validation; Visualization. H.L.: Validation. All authors have read and agreed to the published version of the manuscript.

**Funding:** Major Scientific and Technological Achievements Transformation Project of Ningxia Hui Autonomous Region (2023CJE09053).

**Institutional Review Board Statement:** Not applicable.

**Informed Consent Statement:** Not applicable.

**Data Availability Statement:** The data presented in this study are available on request from the corresponding author.

**Conflicts of Interest:** Author Jintao Zheng was employed by the company Shanxi Shanbei Qianyuan Energy and Chemical Co., Ltd. The remaining authors declare that the research was conducted in the absence of any commercial or financial relationships that could be construed as a potential conflict of interest.

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
