# Peer review of "Improving Physical and Chemical Properties of Saline Soils with Fly Ash Saline and Alkaline Amendment Materials"

_sustainability, doi:10.3390/su16083216_

Round 1

Reviewer 1 Report

Comments and Suggestions for Authors

The authors used fly ash as a chemical amendment to test if it improves the soil physical and chemical properties of saline soil. Moreover, they also tested how the growth parameter of Arrhenatherum elatius in potting condition and  Zea mays L. in field conditions are influenced by the amendment. Generally the manuscript sounds scientific, introduction and discussion sections are clear. Methods section needs some certification. Results are presented well. I have some general comments.

The most of the part of the manuscript doesn't include the values of standard deviation (SD) or standard error (SE) after the results values. So please include it.

In the discussion section its better to indicate figure number where you have been writing about your results.

Also there are many minor grammar mistakes all over the manuscript. Please correct it.

The specific comments are:

Line 1: Better to replace MH with fly ash in the title because the title should be clear.

Line 13-14: Please correct the sentence “plant of Arrhenatherum elatius at room temperature growth conditions were evaluated” by growth conditions of Arrhenatherum elatius at room temperature were evaluated.

Line 15: Please check if there is dot After L. It should be like Zea mays L. And correct it all over the manuscript.

Line 65: Please rewrite the sentence.

Line 80: Please write in italic if it is scientific name.

Line 102-103: Please be specific on what parameter of those plant species will be evaluated.

Line 118- 144: Please clarify the essence of doing the experiment without (section 2.2.1 soil pot cultivation experiment) and with the plants (section 2.2.2. Arrhenatherum elatius planting pot experiments). so that the readers won’t be confused otherwise it sounds confusing.

Line 133: Please rewrite the heading so that it becomes clear.

Line 340-341: Please correct the grammar of the sentence.

Line 342: Please correct the grammar of the sentence.

Line 359: Please correct the grammar of the sentence.

Line 398: Please correct the grammar of the sentence. There are many grammatical mistakes all over the manuscript. Please correct all those.

Comments on the Quality of English Language

There are many minor grammar mistakes all over the manuscript. Please correct it.

Author Response

Dear editor and reviewer:

Thank you very much for your very constructive comments on our manuscript! We have read your comments carefully and have made revision which marked in yellow in the paper. We have tried our best to revise our manuscript according to the comments. Attached please find the revised version, which we would like to submit for your kind consideration.

Thank you very much for your valuable comments. In the text, we have indicated the standard errors used in the data, such as line 301 and line 365. The discussion section follows the order of the experimental results section, so the figure numbers are not indicated.

Q1: Line 1: Better to replace MH with fly ash in the title because the title should be clear.

A1: Thank you very much for your very constructive comments. We replace MH in the text with fly ash, see Line3.

Q2: Line 13-14: Please correct the sentence “plant of Arrhenatherum elatius at room temperature growth conditions were evaluated” by growth conditions of Arrhenatherum elatius at room temperature were evaluated.

A2: Thank you very much for pointing out our shortcomings. We have reworded the sentence, see line 13.

Q3: Line 15: Please check if there is dot After L. It should be like Zea mays L. And correct it all over the manuscript.

A3: We apologize for our error. We rechecked all names in the text and highlighted them in yellow.

Q4: Line 65: Please rewrite the sentence.

A4: Thank you for your advice. We have rewritten the sentence, see line 64.

Q5: Line 80: Please write in italic if it is scientific name.

A5: Thank you very much for your careful review of our article. The middle name of the text has been italicized; please see line 80.

Q6: Line 102-103: Please be specific on what parameter of those plant species will be evaluated.

A6: We have rewritten this sentence based on your suggestion, We have added agronomic indicators for the plants measured, see line 100.

Q7: Line 118- 144: Please clarify the essence of doing the experiment without (section 2.2.1 soil pot cultivation experiment) and with the plants (section 2.2.2. Arrhenatherum elatius planting pot experiments). so that the readers won’t be confused otherwise it sounds confusing.

A7: We gratefully appreciate for your valuable suggestion. We have reorganized this section, see lines 126 to 155.

Q8: Line 133: Please rewrite the heading so that it becomes clear.

A8: Thanks again for your advice. We have rewritten this table heading, see line 144.

Q9: Line 340-341: Please correct the grammar of the sentence.

A9: We have rewritten this sentence based on your suggestion. We checked the grammar of the sentence and rewrote it, see line 365.

Q10: Line 359: Please correct the grammar of the sentence.

A10: Many thanks to your suggestions, We checked the grammar of the sentence and rewrote it, see line 365.

Q11: Line 398: Please correct the grammar of the sentence. There are many grammatical mistakes all over the manuscript. Please correct all those.

A11: Thank you for underlining this deficiency. We checked the grammar of the sentence and rewrote it, see line 421.

We would like to thank the referee again for taking the time to review our manuscript. I am looking forward to hearing from you.

Yours Sincerely,

Changcong An

North Minzu University, China.

Reviewer 2 Report

Comments and Suggestions for Authors

General comments: The overall structure of the manuscript is good. The manuscript possesses promising findings; however, it needs some revisions. One of the main comments is how and why this work is novel and different from others. Articulate how this research contributes to new knowledge in the discipline at the end of the introduction. 

Specific comments:

  • The introduction looks good, but there is a scope to include some texts about heavy metals and how salinity and alkalinity materials affect them.
  • The Methodology section needs a little attention—for example, Subheadings 2.2.1 to 2.2.3. Gradients (7) have been duplicated in many places. These can be placed one and then referred onwards, appropriately.  
  • The results section looks good, but some of the results are duplicated in the discussion sections.  
  • In the conclusions section, the authors can consider adding a general sentence, say…. "Although MH functioning has been proven to show a positive effect on soil and crop, the effect depends on the sources (difference in coal combustion and sintering temp), land use, weather condition, etc." (use better words). The authors can also consider adding gaps/limitations of the current study and future research needs. 
  • Please see a few comments on the manuscript.

Author Response

Dear editor and reviewer:

Thank you for your decision and constructive comments on my manuscript. I have learned a lot from your valuable advice and improved my professional skills. This will be a valuable asset in my future study. We are very sorry for the error in the previous manuscript. We have read your comments carefully and made changes. And here we list the changes and marked in green in revised paper. The specific changes are as follows:

Q1:Line11 define MH here, pls

A1: Thank you very much for your very constructive comments. We have rewritten the sentence, see line 11.

Q2:Line33 Add a ref pls.

A2: Based on your valuable advice, we have added references, see line 32.

Q3:Line42 Add a ref pls. I would suggest to include citation for any data used in the manuscript.

A3: Based on your valuable advice, we have added references, see line 42.

Q4:Line52 Are you sure there are only four?

A4: Thank you very much for your valuable comments on our manuscript. There may be many ways to improve it at the moment, but it can be categorized under these four improvements.

Q5:Line95 Move this part after MH saline

A5: Thank you for your careful reading of our manuscript. We rewrote the words,see line 93.

Q6: Line Did you mean BD?

A6: Thank you for your helpful advice. We have rechecked the content. Please see line 96.

Q7:Line115 Can you t critical vaules in different row for these properties?

A7: We sincerely appreciate the valuable comments. We have rewritten this section, see lines 115-124.

Q8:Line121 Name the lab or departmental lab?

A8: Thank you for your helpful advice. We have added the specific name of the lab, see line 127.

Q9:Line123 Pls explain what does wt% mean for the reader to well understand.

A9: We totally agree with your opinion. We added the meaning of wt%, see line 130.

Q10:Line134 When? April to October 2023?

A10: Many thanks to your suggestions. We have added an experimental time component, see line 147.

Q11:Line135 Duplicated. Can you use only one time and then refer them accordingly?

A11: Many thanks to your suggestions, we have reorganized this section, see line 149.

Q12:Line142 Earlier it was mentioned 9 replications!

A12: Many thanks to your suggestions, we have reorganized this section, see line 151.

Q13: Line153 Please report about fertilizer and pest management for both expt.

A13: Many thanks to your suggestions. However, since our formulations are in the experimental stage and patents are not disclosed, we do not disclose the process of use. The specific operation of the experiment is entrusted to a third party to manage.

Q14:Line159 Namr them here, pls.

A14: Thank you very much for your very constructive comments. We have added the names of heavy metals, see line 170.

Q15:Line242 Atre you refereing about soils?

A15: Thank you very much for pointing out our shortcomings. We have rewritten this section, see line 256.

Q16:BD? Can you use IS unit, say, Mg/m3

A16: Thanks for pointing out the problem. We have consulted numerous texts and found that they are all expressed as g/cm3, so the original text is expressed in this form.

Q17:Line275 For the first time you used here!

A17: Thank you very much for your careful review of our article. We have added the names of heavy metals, see line 170.

Q18:Line320 Pls double check the data?

A18: Thank you for your advice. We scrutinized the experimental data.

Q19:Line398 Earlier you have used weight!

A19: We apologize for our error, we have corrected the text, see line 421.

Q20:Line617 Tis part should move to the results section.

A20: Many thanks to your suggestions, we moved this section to the results section, see line 494.

Q21:Line652 Put a general statement not results.

A21: Thanks again for your advice, we are adjusting this part of the presentation based on your suggestion, please see line 674.

Q22:Line666 Reword. Say e.g., Although MH functioning has been proven to show positive effect on soil and crop but the effect depends on the soures (difference in coal combustion and sintering temp), land use, weather condition etc.

A22: We gratefully appreciate for your valuable suggestion. We have reorganized this section based on your valuable suggestions, please see line 683.

We would like to thank the referee again for taking the time to review our manuscript. I am looking forward to hearing from you.

Yours Sincerely,

Changcong An

North Minzu University, China.

Reviewer 3 Report

Comments and Suggestions for Authors

The authors examined the use of ornamental plants and fly ash in the improvement of saline soils. According to the results, the methodology used was efficient. However, to improve the manuscript, some aspects need to be revised.

-Title. Please, replace MH to “fly ash”.

-Excessive references were used.

-More photos of the experimental plan are needed

-Photo of each plant mentioned in the text must be inserted

-The manuscript is not suitable according to the journal template. E.g. font, titles, subtitles, abbreviations, symbols, figure caption and legends. Review everything.

-If the methodology used is to be extended to areas with this type of soil, is it not necessary to show a map with the areas of saline soil in the region?

-The manuscript needs a flowchart to understand the methodology. It also needs a flowchart to understand the soil improvement process with the method used.

Author Response

Dear editor and reviewer:

On behalf of my co-authors, thank you for your careful review of our manuscript and your very constructive suggestions! We thank you for giving us a chance to revise and improve the quality of our article.

We have read your comments carefully. We have tried our best to revise our manuscript according to the comments. Attached please find the revised version, which we would like to submit for your kind consideration.

Q1: Title. Please, replace MH to “fly ash”.

A1: Thank you very much for your very constructive comments. We replace MH in the text with fly ash, see Line3.

Q2: Excessive references were used.

A2: Thank you very much for your valuable comments on our manuscript. We would like to attribute any data we use. I don't know if that's feasible.

Q3: More photos of the experimental plan are needed

A3: Thank you for your helpful advice. However, we have not added pictures of the process of using the formula, as it is still in the experimental stage and the patent has not been disclosed.

Q4: Photo of each plant mentioned in the text must be inserted

A4: We sincerely appreciate the valuable comments. We really apologize for the fact that due to the progress of the experiments carried out, the early pictures of the plants were not retained, only the data related to the plants were kept.

Q5: The manuscript is not suitable according to the journal template. E.g. font, titles, subtitles, abbreviations, symbols, figure caption and legends. Review everything.

A5: I'm very sorry for our mistake. We totally agree with your opinion. We have corrected everything in the text according to the template, including the figure notes, etc.

Q6: If the methodology used is to be extended to areas with this type of soil, is it not necessary to show a map with the areas of saline soil in the region?

A6: Many thanks to your suggestions. Since we are still in the trial stage, we will subsequently consider doing a further extension stage. Realize wide applicability.

Q7: The manuscript needs a flowchart to understand the methodology. It also needs a flowchart to understand the soil improvement process with the method used.

A7: Thank you very much for your very constructive comments. We have added a design chart for the experiment, see line 140.

We would like to thank the referee again for taking the time to review our manuscript. I am looking forward to hearing from you.

Yours Sincerely,

Changcong An

North Minzu University, China.

Round 2

Reviewer 1 Report

Comments and Suggestions for Authors

Please check the tables all over the manuscripts, they are not clear. other portion of the manuscript is fine.

Comments on the Quality of English Language

English can be improved.

Reviewer 3 Report

Comments and Suggestions for Authors

I agree to publish